# LINC00892 Is an lncRNA Induced by T Cell Activation and Expressed by Follicular Lymphoma-Resident T Helper Cells

**DOI:** 10.3390/ncrna8030040

**Published:** 2022-06-01

**Authors:** Ingram Iaccarino, Fatme Mourtada, Sarah Reinke, Paurnima Patil, Gero Doose, Gianni Monaco, Steve Hoffmann, Reiner Siebert, Wolfram Klapper

**Affiliations:** 1Department of Pathology, Hematopathology Section and Lymph Node Registry, University of Kiel, 24105 Kiel, Germany; fmourtada@fz-borstel.de (F.M.); sreinke@path.uni-kiel.de (S.R.); wklapper@path.uni-kiel.de (W.K.); 2Institute of Genetics and Biophysics “A. Buzzati-Traverso”, Consiglio Nazionale delle Ricerche, 80131 Naples, Italy; 3Institute of Human Genetics, Ulm University, Ulm University Medical Center, 89081 Ulm, Germany; paurnima.patil@uni-ulm.de (P.P.); reiner.siebert@uni-ulm.de (R.S.); 4ecSeq Bioinformatics GmbH, 04103 Leipzig, Germany; gero.doose@ecseq.com; 5Institute for Transfusion Medicine and Gene Therapy, Medical Center—University of Freiburg, 79106 Freiburg, Germany; gianni.monaco@uniklinik-freiburg.de; 6Leibniz Institute on Aging—Fritz Lipmann Institute (FLI), 07745 Jena, Germany; steve.hoffmann@leibniz-fli.de

**Keywords:** lncRNA, LINC00892, T cell activation, PD1, T helper cells, Follicular Lymphoma

## Abstract

Successful immunotherapy in both solid tumors and in hematological malignancies relies on the ability of T lymphocytes to infiltrate the cancer tissue and mount an immune response against the tumor. Biomarkers able to discern the amount and the types of T lymphocytes infiltrating a given tumor therefore have high diagnostic and prognostic value. Given that lncRNAs are known to have a highly cell-type-specific expression pattern, we searched for lncRNAs specifically expressed by activated T cells and at the same time in a kind of lymphoma, follicular lymphoma, where the microenvironment is known to play a critical role in the regulation of antitumor immunity. We focused on a non-coding transcript, annotated as LINC00892, which reaches extremely high expression levels following cell activation in Jurkat cells. Interestingly LINC00892 has an expression pattern resembling that of genes involved in T cell memory. Accordingly, LINC00892 is mostly expressed by the effector memory and helper CD4+ T cell sub-types but not by naïve T cells. In situ analyses of LINC00892 expression in normal lymph nodes and in follicular lymphoma biopsies show that its expression is limited to CD4+ PD1^hi^ T cells, with a subcellular localization within the germinal center matching that of follicular helper T cells. Our analysis therefore suggests that the previously uncharacterized lncRNA LINC00892 could be a useful biomarker for the detection of CD4+ memory T cells in both normal and tumor tissues.

## 1. Introduction

Cancer microenvironments play a crucial role in several aspects of tumor development [1,2]. Among the different cell types that shape and influence the tumor microenvironment, infiltrating T lymphocytes are those that have the strongest therapeutic potential, being able to counteract and possibly eradicate tumor growth by mounting an adaptive immune response against the tumor itself [3]. At the same time, a quantitative assessment of intratumoral T cell content has no prognostic value if not associated with a qualitative definition of the specific T cell subset present in the tissue. It is known, for instance, that the presence of intratumoral memory T cells is associated with good outcomes [4], but worse outcomes were observed in patients with an increased number of infiltrating CD4+ Tregs [5].

Non-Hodgkin B-cell lymphomas (NHBL) mostly derive from lymph-node-resident B cells that undergo uncontrolled cell growth following a process of oncogenic transformation. In normal lymph nodes, T cells play a crucial physiological role in the development of the immunoglobulin-mediated humoral immune response. By establishing cell-to-cell contacts with B cells, CD4+ follicular helper T cells (TFH cells) improve the survival of antigen-recognizing B cells and induce immunoglobulin class switching. Follicular lymphoma (FL) is an NHBL where the follicular structure of a normal lymph node is mostly conserved and where the microenvironment is known to play an important role in tumor progression [6,7]. Although mostly considered indolent, FL often relapses and can transform to a very aggressive disease with poor overall survival [8]. Several studies have shown that a reactive microenvironment as well as specific T cell subpopulations, including tumor-infiltrating programmed cell death 1 (PD1) positive lymphocytes, could play a major role in FL clinical behavior [9,10,11]. Within FL tissue, T cells appear to have a nonrandom distribution relative to malignant B cells and to be organized in clusters [12]. Recently, specific populations of CD4+ memory T cells were also associated with FL survival [13]. The ability to distinguish between the several sub-types of T cells that infiltrate a given cancer tissue at a given progression stage therefore has crucial prognostic importance, particularly in FL.

The development of novel sequencing technologies has been accompanied by a tremendous increase in our knowledge of the human genome and transcriptome. It is now clear that the human genome is able to produce at least as much RNA without protein coding potential as RNA coding for proteins. Among all non-coding transcripts present in human cells, those classified as long non-coding RNAs (lncRNAs) are emerging as crucial players in the control of gene expression [14,15,16]. By definition, lncRNAs are RNAs longer than 200 nucleotides and with no predicted coding potential. They can be divided in several subclasses according to their location relative to coding genes (intergenic, intronic or overlapping with exons of coding genes in antisense or sense orientations) [17,18,19]. In an attempt to characterize the expression of this critically understudied class of RNAs, one of the general features that emerged was that lncRNAs exhibit more specific expression patterns than coding RNAs, meaning that there are more lncRNAs than protein-coding genes specifically expressed in a single cell type or tissue type [17,18,19]. LncRNAs have also the tendency to have expression patterns specific for cancer sub-types and disease stages [20,21]. These features make lncRNAs ideal targets for cancer biomarker discovery [22].

With the aim of identifying novel markers for NHBL-resident activated or memory T cells, we performed transcriptomic analysis of a model T cell line acquiring a status similar to that of memory T cells and intersected the data with the expression data produced by the ICGC-MMML-seq project on lymphoma specimens from 3 major types of germinal-center-derived B-cell lymphoma [23,24]. With this approach, we identified several lncRNAs strongly expressed in FL samples and in a cell line with acquired features of memory T cells. We present here data on the characterization of LINC00892, a long non-coding RNA transcript expressed by tissue-resident TFH cells that is strongly induced and persistently expressed following T cell activation.

## 2. Results

### 2.1. Identification of lncRNAs Expressed by FL-Resident T Cells

In order to identify lncRNAs expressed by B-cell lymphoma-resident memory T cells, we set up an experimental pipeline where transcriptomics data from non-Hodgkin B-cell lymphomas were compared with gene expression data from Jurkat cells being stimulated to acquire a “memory-like” status as described previously [25,26,27]. Details on this experimental system are presented in Appendix A. A list of lncRNAs differentially expressed in memory-like Jurkat cells (Appendix A) was compared with lncRNAs differentially expressed in follicular lymphoma (FL) [21]. We focused on FL because is a type of lymphoma particularly rich in stroma and where the microenvironment has been shown to play an important role in tumor progression [6,28].

Figure 1A shows a mean-difference (MD) plot of the highest expressed and most differentially regulated lncRNAs in the transition between non-stimulated to memory-like Jurkat cells. Transcripts similarly regulated in FL are highlighted in red. A list of lncRNAs regulated in the same direction both in Jurkat cells and in FL is presented in Appendix A.

In order to select for T cell-specific transcripts, we interrogated several public repositories for the expression of some of the most significantly regulated lncRNAs. Analysis of data from the Blueprint consortium in the Expression Atlas at the EBI-EMBL website shows that LINC00892 had an expression pattern uniquely limited to CD4+ and CD8+ memory T cells (Figure 1B). T-regulatory cells also expressed LINC00892, although at lower levels compared to memory CD4+ T cells. On the contrary, both LINC00963 and LINC00324 had broader patterns of expression in the hematopoietic lineage (Figure 1B). MIR223HG, the host gene of hsa-mir223, although strongly induced in both Jurkat cells and FLs, was shown to be more specific for the myeloid lineage [30]. Restricted T cell expression of LINC00892 was evident also in the data from Monaco et al. [31], showing strong expression in most of the differentiated CD4+ T cell lineages (Appendix A).

Calderon and co-workers performed a comprehensive RNA sequencing analysis of up to 32 immune cell populations under resting conditions and after stimulation [29]. In order to understand whether the LINC00892 gene is induced by TCR stimulation also in primary T cell subgroups, we reanalyzed the publicly available data sets and looked at the expression of LINC00892 in naïve CD4+ T cells and in differentiated CD4+ memory T cell subgroups treated or not with anti-CD3/CD28 beads for 24 h. The data is presented in Figure 1C and shows that in naïve CD4+ T cells, TCR stimulation induced a strong and significant upregulation of LINC00892 (log Fold change = 6.33); in all memory T cell subsets, where LINC00892 expression is already high (in agreement with both Blueprint and Monaco et al. [31]), TCR stimulation induces further increases in LINC00892 expression, reaching statistical significance in some of the subsets. LINC00892 expression is therefore strongly induced by TCR stimulation not only in the acute T cell leukemia Jurkat cell line but also in primary T cells from healthy individuals.

### 2.2. Genomic Characterization of LINC00892

The LINC00892 gene is located at band q26 of chromosome X. LINC00892 has a 3-exon structure and is transcribed on the + stand of the genome, approximately 6 kb upstream of the CD40LG gene. Concordance between the gene structure and the mapped reads from both Jurkat cells and lymphoma samples is shown in Figure 2A. Furthermore, cloning of the LINC00892 gene from a cDNA amplified from an FL sample revealed the existence of an un-annotated transcript isoform missing the second exon (Appendix A). According to both the Coding Potential Calculator (CPC2, http://cpc2.cbi.pku.edu.cn/index.php; accessed on 19 April 2022) and the Coding Potential Assessment Tool (CPAT, http://lilab.research.bcm.edu/cpat/index.php; accessed on 19 April 2022) LINC00892 is classified as a noncoding sequence with coding probability around 0.01. Subcellular fractionation studies in activated Jurkat cells show that LINC00892 is more abundant in the cytosolic fraction relative to the nuclear fraction (Appendix A). With few exceptions, lncRNAs are known to be poorly conserved at the sequence level [32,33]. Searching for a mouse LINC00892 homologue, we analyzed the region upstream of the Cd40lg gene in the *mus musculus* genome (mm9) and noticed the presence of a bi-exonic transcript (Gm14718) annotated with the biotype lncRNA at a distance from the Cd40lg gene (7.5 kb) very similar to that observed in the human genome between LINC00892 and CD40LG (Figure 2B). Interestingly, by aligning the two sequences, we found a significant degree of sequence homology, particularly in the 5′ and 3′ regions of the transcribed sequences (Figure 2C). According to the Expression Atlas, Gm14718 also has a T cell-restricted expression pattern (Appendix A). In order to investigate if Gm14718 could be a functional homologue of LINC00892, we examined whether Gm14718 is also induced by TCR stimulation in CD4+ mouse T cells. To this purpose, we reanalyzed publicly available RNA-seq data obtained from mouse CD4+T cells treated or not with CD3/CD28 antibodies (GEO Dataset: GSE34550) [34] and found that Gm14718 was strongly induced following CD3/CD28 treatment (Figure 2D). This result clearly suggests that Gm14718 is a bona fide functional mouse homologue of LINC00892.

### 2.3. Expression of LINC00892 in Normal and Neoplastic Tissues

The expression data from normal hematopoietic cells show that LINC00892 is not expressed in B cells (Figure 1B). The high LINC00892 expression found in FL tumor samples could either mean that transformed B cells have acquired the capability to express LINC00892 or that LINC00892 is expressed by tumor-resident T cells. To address this issue, we performed RT-qPCR analysis of LINC00892 expression in CD4+ T cells, CD8+ T cells and B cells sorted from an advanced FL case. We also analyzed in parallel material from a patient affected by lymph node hyperplasia (LN). As Figure 3 clearly shows, although expressed also in CD8+ T cells, LINC00892 was found expressed mainly in tissue-resident CD4+ T cells. No expression was found in B cells.

The expression analyses shown in Figure 1 and Appendix A were performed on cells sorted by FACS from mononuclear cells derived from peripheral blood (PBMC). The data from Figure 3 suggest that LINC00892 is expressed also from tissue-resident T cells, either in a non-neoplastic (but reactive) tissue (LN) or from a neoplastic tissue (FL). Indeed, analysis of data from the Genotype-Tissue Expression (GTEx) Project shows that besides blood, LINC00892 is expressed also in spleen, lung and small intestine Peyer’s patches (Appendix A). LINC00892 expression seems therefore to be specific to body sites involved in an active immune response, usually orchestrated by germinal centers.

In line with an expression specific to cell components of germinal centers, when we analyzed LINC00892 expression in the pan-cancer data produced from the International Cancer Genome Consortium (ICGC) [35], we found that among the different cancers, LINC00892 expression was particularly high in FL samples, a lymphoma where the structure of the germinal center is particularly conserved (Appendix A).

### 2.4. In Situ Expression Analysis of LINC00892 in Tumor Samples

Among the different T cell types present in germinal centers, follicular helper T cells (TFH) are a specialized subset of CD4+ T cells that plays a critical role in adaptive immunity, supporting B cells in the development of antibodies against foreign pathogens. Interestingly, by looking at genes that correlate with LINC00892 expression in the data from Monaco et al. [31], we found that genes like CCR4, CD40LG, ICOS and PDCD1, all typically expressed by TFH cells, had a very high positive expression correlation with LINC00892 (Appendix A). In order to investigate whether LINC00892 is indeed expressed by activated TFH cells, we used in situ hybridization (ISH) detection in formalin-fixed paraffin-embedded (FFPE) tissue slides from LN and FL samples. ISH experiments were performed using a LINC00892-specific probe based on the RNAscope technology [36]. As shown in Figure 4, ISH analysis revealed a very specific spot-like staining associated with a defined cellular population in both LN and the FL tissues. Focusing on germinal centers in the LN tissue, we found that LINC00892 was expressed by a subpopulation of CD4+ T cells more abundant in the light zone and with a strong positivity for PD1 expression. A similar association with PD1 was observed also in FL samples (Figure 4E,F). This analysis is therefore a strong suggestion that in both LN and FL, LINC00892 is expressed by TFH cells.

### 2.5. Single-Cell Analysis of LINC00892 Expression in Jurkat Cells

Relative to RT-qPCR, ISH has the advantage of enabling expression analysis in single cells. We therefore also performed ISH with the LINC00892-specific probe in Jurkat cells treated or not with PMA/ionomycin for 48 h. Visual inspection of the results showed that in both treated and untreated samples, not all of the cells were positive for LINC00892 expression (Figure 5A,B). ISH analysis using RNAscope reagents produces signal spots that are derived from the amplification of single RNA molecules. In order to quantify LINC00892 expression in treated and untreated Jurkat cells, we performed computer-assisted image analysis. Image fields were analyzed to count the number of cells having no spots, those having few spots and those having many spots. Interestingly, in untreated samples, almost 90% of the cells had no spots specific for LINC00892 (Figure 5D,E). This was not due to a technical artifact, as a probe recognizing Cyclophilin B (PPIB) stainined more than 80% of the cells (Figure 5C,D). In PMA/ionomycin-treated samples, the number of LINC00892-negative cells was still above 60%. In order to understand if the strong difference in LINC00892 expression observed in the Jurkat cell population was due to cellular heterogeneity of the Jurkat cell line used, we sub-cloned the cell line and repeated the analysis in a single-cell clone (Clone AC2). Interestingly, although Clone AC2 showed a mild increase in cells with spots, it was still characterized by strong cellular heterogeneity, with more than 80% of the cells expressing no LINC00892 in untreated conditions (Figure 5D). Contrary to the parental cell line, treatment with PMA/ionomycin increased the number of cells positive for LINC00892 to 70%. The same kind of picture is evident if we analyze the number of spots per cell in all conditions as a measure of intensity of LINC00892 expression (Figure 5E). This observation suggests that Jurkat cells might be heterogeneous in their ability to respond to PMA/ionomycin in the activation of LINC00892 expression. Interestingly, a similar heterogeneous expression pattern has been observed previously by ISH in the analysis of cytokine expression in cloned Th1, Th2 and Th0 cell lines [37].

### 2.6. Characterization of LINC00892 Expression in Jurkat Cells

As shown in Figure 2A, LINC00892 is strongly induced in Jurkat cells following treatment with a combination of PMA and ionomycin. The reason for the concomitant use of PMA and ionomycin to induce T cell activation is based on the need to activate multiple signal transduction pathways. PMA is a potent activator of the PKC pathway, while ionomycin, by increasing intracellular calcium levels, induces the activation of NFAT. Genes like IL2 typically require both drugs for full transcriptional activation, but for the activation of other genes, the presence of either PMA or ionomycin might be sufficient [38]. In order to characterize the details of LINC00892 expression following Jurkat cell activation, we performed several analyses. First, we analyzed the kinetics of LINC00892 expression following PMA/ionomycin treatment; second, we investigated whether transcriptional induction required the concomitant presence of PMA and ionomycin; third, we determined to what extent the intensity and duration of the stimulus influenced LINC00892 expression. As shown in Figure 6A, we found that, contrary to IL2 (Appendix A), LINC00892 expression was late and persistent: following PMA and ionomycin treatment, its expression barely changed after 3 h, started to increase after 24 h and reached its maximum after 72 h. A second stimulation 4 days after stimulus withdrawal still increased LINC00892 expression but without reaching statistical significance. The expression behavior of LINC00892 in Jurkat cells is therefore strikingly similar to that observed in naïve and memory primary T cells (Figure 1C). We also found that most of the increase in LINC00892 expression after stimulation was due to the action of PMA only. The use of ionomycin alone did not vary LINC00892 expression, and the concomitant use of PMA and ionomycin resulted in a further but not significant increase relative to PMA alone (Figure 6B). Next, we tested several concentrations and stimuli durations and found that PMA at 1 ng/mL given for 5 min was already able to induce a strong increase in LINC00892 expression (Figure 6C). As shown in Appendix A, PMA and ionomycin treatment was accompanied by cell growth arrest with a strong induction of CDKN1A, the gene coding for the cell cycle inhibitor p21. As for LINC00892, CDKN1A induction was also dependent on PMA only (Appendix A). Interestingly, PMA has already been shown to induce p21 expression and cell cycle arrest in other cell systems [39]. Given that LINC00892 expression in our experimental conditions was associated with p21 induction and growth arrest, we asked if other stimuli that induce cell cycle arrest and p21 induction would result in an increase in LINC00892 expression. As shown in Figure 6D, serum starvation of Jurkat cells for 48 h, although associated with a strong induction of CDKN1A, was not accompanied by an increase in LINC00892 expression.

### 2.7. RNA Interference of LINC00892 in Jurkat Cells

As mentioned above, the LINC00892 gene is located 6 kb upstream of the CD40LG gene. CD40LG is a protein involved in several aspects of the immune response and is known to be tightly regulated [40]. Looking at publicly available expression data, the CD40LG gene has an expression pattern very similar to that of LINC00892 in PBMC ([31] and Appendix A). Furthermore, data from the Pan-Cancer Analysis of Whole Genomes (PCAWG) study on the UCSC Xena browser (https://xena.ucsc.edu/; accessed on 19 April 2022) shows a very high positive expression correlation between LINC00892 and CD40LG (Figure 7A). This expression correlation could result either from the presence of common regulatory elements controlling the expression of the two genes or from a positive regulation of one of the transcripts over the other. Particularly, we tested the hypothesis that LINC00892 could regulate CD40LG expression, given that other lncRNAs have been shown to be able to control the expression levels of coding genes [16,41]. To test this hypothesis, we performed RNAi of LINC00892 in Jurkat cells and analyzed CD40LG expression by RT-qPCR. As shown in Figure 7B, the use of siRNAs directed against LINC00892 successfully knocked down both basal and PMA-induced LINC00892 expression but induced no change in CD40LG gene expression. We therefore performed whole transcriptome analysis using RNA-seq, comparing PMA-treated Jurkat cells transfected either with a control or with a LINC00892-targeting siRNA pool. Figure 7C,D shows that besides LINC00892, only 3 genes were found significantly differentially expressed after LINC00892 knock-down. Therefore, reduction of LINC00892 expression after PMA treatment in Jurkat cells in the investigated conditions has no major effect on the cellular transcriptome.

## 3. Discussion

The identification of RNA transcripts with a pattern of expression specific for particular cell types and activation statuses can be extremely useful for improving the prognostic and diagnostic power of gene expression signatures. This holds particularly true in the analysis of tumor microenvironments, where the quantity and quality of tissue-infiltrating lymphocytes can have predictive importance in patients undergoing immunotherapy. Given that memory T cells have been associated with better overall survival in several cancers and given that lncRNAs have been shown to have high cell-specific expression, we searched for lncRNAs expressed at high levels in memory T cells and at the same time in sequencing data from different B-cell lymphomas. We focused on a transcript, LINC00892, with high expression both in follicular lymphoma (FL) and in a cell system resembling activated memory T cells. By mining public RNA-seq data on several PBMC purified components, we could confirm that LINC00892 is exclusively expressed in T cells that underwent differentiation after cellular activation. The highest expression values can be observed in CD4+ T cells in general, with particularly high expression in the effector memory CD4+ and the T helper sub-types. Interestingly, both CD4+ and CD8+ naïve T cells show no expression of LINC00892, but once treated with stimuli that trigger TCR signaling, they also increase LINC00892 expression (Figure 1C). These observations are very well recapitulated in our Jurkat cellular model. We observe a strong increase in LINC00892 expression in Jurkat cells after treatment with a 5 min pulse of PMA. Interestingly, contrary to genes like IL2 whose expression is only transient after stimulation (Appendix A), LINC00892 expression persists at a very high level for several days after stimulation (Figure 6A). Therefore, LINC00892 has a pattern of expression similar to typical memory genes, such as those shown in Appendix A. Interestingly, single-cell expression analysis showed that LINC00892 has a strong heterogeneous expression pattern in Jurkat cells (Figure 5). This observed heterogeneous expression in single cells of a clonal cell population may be the result of asynchronous transcriptional bursts accompanied by rapid RNA degradation. Although we did not measure the half-life of the LINC00892 RNA, we did notice that upon ectopic expression in Jurkat cells, high levels of LINC00892 expression lasted only 24 h (data not shown). On the other hand, a similar heterogeneous expression behavior has been observed previously in the analysis of cytokine mRNAs in cloned Th1, Th2 and Th0 cell lines [37]. Here, the authors suggest that even in clonal T cell populations, there may be a high level of plasticity in the expression of cytokine genes. In line with this hypothesis, we find that a Jurkat sub-clone (AC2) is more homogeneous in its ability to increase LINC00892 expression (Figure 5D) and reaches higher expression levels. Interestingly, Clone AC2 also has an enhanced ability to increase CD40LG expression (data not shown), suggesting that single-cell plasticity may be at the level of transcriptional regulation.

One interesting feature of LINC00892 is that it is located very close to the gene coding for the CD40 ligand, CD40LG. The ligand for CD40 plays a crucial role in the communication between T helper cells and B cells in the context of the germinal center. Indeed, mice in which CD40LG is deleted do not develop germinal centers and have a strong defect in the humoral immunological response due to a failure to undergo immunoglobulin class-switch recombination [40]. Looking at expression data from both normal cells and tumor material, we found a striking expression correlation between LINC00892 and CD40LG (Figure 7A). Among T cells, only naïve T cells do not show co-expression of the two genes. The expression correlation could either imply that the genes are able to regulate each other or that the genes are part of the same transcriptional unit regulated by a common enhancer. To investigate whether LINC00892 had the capability to regulate the expression of CD40LG, we performed RNAi of LINC00892 and looked at CD40LG expression. As we show in Figure 7B, we could not detect any difference in the expression of CD40LG in conditions where the levels of LINC0892 were strongly reduced. We therefore rule out the possibility that LINC00892 could be a transcriptional or post-transcriptional regulator of CD40LG. However, we cannot exclude the possibility that in other cell types or experimental conditions, a reciprocal regulation between the two genes could take place. We postulate therefore that the strong expression correlation between LINC00892 and CD40LG expression might rely on the presence of common regulatory regions upstream of the two genes. The analysis shown in Appendix A indeed suggest that the 30 kb of genome upstream of CD40LG, including LINC00892, has several epigenetic features found in enhancer regions, with segments displaying differential methylation between CD4+ memory T cells and naïve CD4+ T cells, the only cell types showing a discrepancy in co-expression. Interestingly, ChIP-seq data from primary CD4+ T cells [42] suggest that some transcription factors can drive the transcription of two genes concomitantly (ETS1) and others (RUNX1) of one transcriptional unit only.

As shown in Figure 6A, LINC00892 sees a strong increase in expression following PMA treatment. In order to understand whether LINC00892 was able to affect the cellular transcriptome in these conditions, we performed RNA-seq analysis of cells treated with PMA and then either with a non-targeting siRNA pool or with a pool targeting LINC00892. As shown in Figure 7D, the reduction of LINC00892 expression in these conditions had a very modest effect on the cellular transcriptome, with only 4 genes significantly changing in expression level, one of them being LINC00892 itself. Several scenarios could explain this result: (a) LINC00892 function in Jurkat cells, a leukemia cell line, is limited; (b) RNAi was not able to reduce LINC00892 levels enough to allow observation of significant effects; (c) LINC00892 expression may affect the cellular proteome and not the transcriptome; (d) LINC00892 may have a function that is evident only at the organismal level. Performing the right experiments to provide answers to these questions goes beyond the scope of this manuscript. At the same time, our detailed analysis of this novel transcript, with the concomitant finding of a bona fide mouse homologue of LINC00892 (Gm14718), will certainly be a strong incentive for the establishment of mouse models carrying genetic lesions in this locus that will definitely reveal the role of LINC00892 at the organismal level and in the immunological response in particular.

As mentioned above, one of the interesting features of lncRNAs is their specific expression patterns relative to protein coding genes. LINC00892 expression is not only limited to a few cell types but also to their activation state. This makes of LINC00892 a potentially interesting biomarker for activated T cells (Figure 8).

Since the discovery that activated T cells can mount a strong immune reaction against tumors that can lead in some cases to tumor eradication, the identification of the number and the type of tumor-infiltrating T lymphocytes in every single tumor has become an extraordinary predictive tool. In this work, we have shown for the first time that LINC00892 can successfully be detected by ISH in normal lymph nodes as well as in biopsies form a follicular lymphoma patient. Furthermore we could find that this lncRNA is expressed by the same cells expressing high levels of PD1, the checkpoint inhibitor receptor that is currently the most targeted protein in cancer immunotherapy [3]. More work is needed to understand whether LINC00892 is further able to discriminate between several metabolic states of activated CD4+ PD1^hi^ infiltrating T cells. However, the results presented here suggest that LINC00892 is a promising candidate in the development of gene expression signatures for tissue deconvolution analyses as prognostic and diagnostic tools in the fight against cancer.

## 4. Materials and Methods

### 4.1. Cell Culture

The Jurkat cell line Clone 20 was a kind gift of Dr. De Berardinis (Institute of Biochemistry and Cell Biology, CNR, Naples, Italy). Cells were grown in RPMI medium supplemented with 10% fetal bovine serum (FBS) and 1x Glutamine. Cell stimulation was performed using 20 ng/mL of PMA and 500 ng/mL of ionomycin in full medium for the indicated amount of time. For the induction of a “memory-like” status, cells were stimulated for 3 h as above and washed 2 times with full medium. Cells were left to rest in the incubator for 4 days and stimulated again as above (see Appendix A for details). For RNA interference experiments, the day before transfection, cells were treated for 3 h with 10 ng/mL PMA or an equal volume of DMSO, washed twice with 5 mL of medium and left to rest in the incubator. The day after, 1.5 × 10^6^ cells were transfected with 50 pmoles of siRNAs using an Amaxa Nucleofector (Lonza, Basel, Switzerland) using Kit V and program X-001. We used either a non-targeting (NT) siRNA pool (Dharmacon, Lafayette, CO, UAS. Cat. D-001320-10-05) or a LINC00892-targeting siRNA pool (Dharmacon, Cat. R-185159-00-0005). After 30 h, cells were collected by centrifugation, and RNA was extracted as indicated later. The experiment was performed in triplicate.

### 4.2. Gene Expression Analyses

After cell stimulation for the indicated times, cells were collected by centrifugation and resuspended in Buffer RLT (Qiagen, Hilden, Germany) for cell lysis and further processed for total RNA purification using the RNeasy^®^ Mini RNA extraction kit (Qiagen), following manufacturer instructions. For real-time quantitative PCR (RT-qPCR) analysis of gene expression, 800 ng of total RNA were reverse transcribed using the QuantiTect Reverse Transcription kit (Qiagen). RT-qPCR was performed using either the Luminaris Color HiGreen qPCR Master Mix (Thermo Fisher Scientific, Waltham, MA, USA) or the ORA^TM^ SEE qPCR Green Mix (highQu, Karlsruhe, Germany) and run on a Light Cycler 480 (Roche, Basel, Switzerland). A list of primers used is presented as a Appendix A. For RNA sequencing, RNA was quality-checked using a Bioanalyzer (Agilent, Santa Clara, CA, USA) and sent for sequencing either at the sequencing facility of Kiel University or at GENEWIZ Europe. In both cases, a TruSeq stranded mRNA seq library preparation kit (Illumina, San Diego, CA, USA) was used, and samples were sequenced on an Illumina NovaSeq sequencer. For each condition, three technical replicates were analyzed.

### 4.3. In Situ Hybridization and IHC

In situ hybridization was performed using the RNAscope^®^ 2.5 HD Detection Kit—BROWN (ACD, Tokyo. Japan. Cat. No. 322310). A LINC00892-specific probe (ACD, Cat No. 499371) was designed by RNAscope based on the sequence of the LINC00892 long isoform (NR_038461). The protocol was developed exactly as described by the manufacturer using a HybEZ™ Oven. DapB and PPIB were used as negative and positive controls, respectively. When combined with IHC, IHC was started after the DAB reaction.

### 4.4. Data Analysis

RNA-seq data were analyzed using the open-source, web-based Galaxy platform [43]. Paired FastQ files were aligned to the human genome (GRCh38) using HISAT2. Counts files were generated with featureCounts using GRCh38.102.gtf as a gene annotation file. Differential expression was calculated using the limma-voom program on the three technical replicates, filtering out genes with expressions lower than 0.5 CPM. Gene annotation files were downloaded from the Ensembl web site (http://www.ensembl.org; accessed on 19 April 2022). Graphical representation of RNA-seq results was obtained using Instant Clue, an open-source visual analytics software [44]. Image analysis of Jurkat cells stained by ISH as described in Figure 5 was performed using the Definiens Tissue Studio^®^ Software. Significance in gene expression was calculated using a two-tailed Student’s *t*-test. A *p*-value < 0.05 was considered significant.

## Figures and Tables

**Figure 1 ncrna-08-00040-f001:**
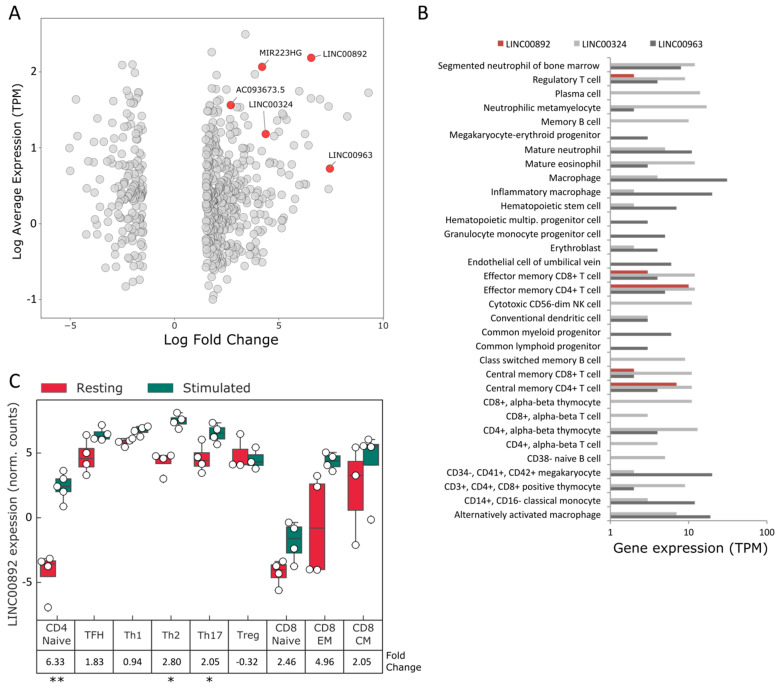
Expression analysis of lncRNAs differentially expressed in Jurkat cells and in FL samples. (**A**) mean-difference (MD) plot of lncRNAs differentially expressed in the comparison of NS with SW4D Jurkat cells. The graph highlights the fold changes in expression of lncRNAs with high expression level (defined in TPM). LncRNAs also differentially expressed in the same direction in FL samples are depicted in red and labeled with their names. (**B**) Gene expression analysis of the three indicated lncRNAs in cells from peripheral blood. The analysis has been performed on data provided by the Blueprint project and downloaded from the Expression Atlas portal of the European Bioinfomatics Institute. (**C**) Analysis of LINC00892 expression in resting conditions and 24 h after treatment with anti-CD3/CD28 beads (stimulated) in different subsets of primary T cells, using the RNA-seq data published by Calderon and colleagues [29]. LINC00892 expression is shown as normalized counts. For each cell line, the log of the fold change (in the comparison of stimulated vs. resting) is shown. Expression analysis has been performed using data from 4 different samples. * *p*-value < 0.05; ** *p*-value < 0.01.

**Figure 2 ncrna-08-00040-f002:**
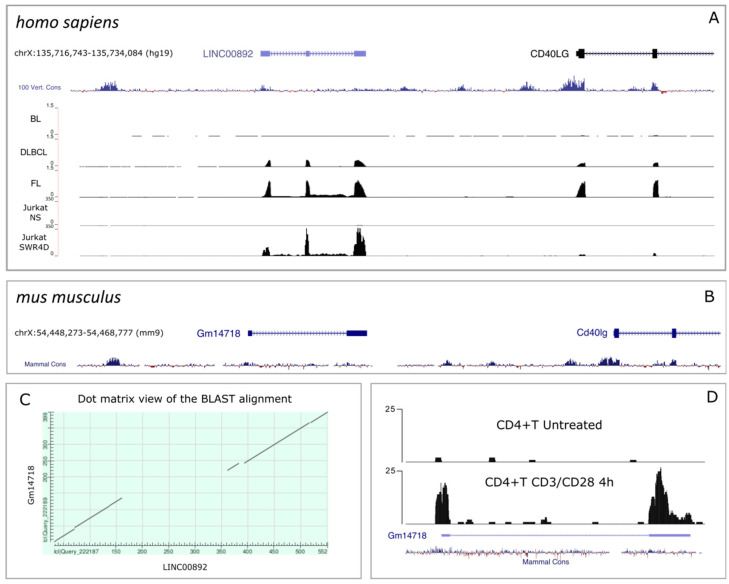
Genomic analysis of LINC00892 and its bona fide mouse homologue. (**A**) Alignment of RNA-seq reads from lymphoma samples (BL, DLBCL and FL, as in [21]) as well as non-stimulated Jurkat cells (NS) and Jurkat cells grown for 4 days after stimulus withdrawal (SW4D). The alignment was performed with the UCSC genome browser on the human genome (h19 assembly) and shows the gene annotation, the chromosome coordinates and the degree of sequence conservation among vertebrates. (**B**) UCSC genome browser graphic of the genomic region of the mm10 assembly of the mouse genome, including the lncRNA Gm14718 and part of the gene coding for Cd40lg. The graph shows the chromosome coordinates and the degree of sequence conservation among mammals. (**C**) Dot matrix view of the alignment between the LINC00892 and the Gm14718 transcripts using the BLAST program. (**D**) Alignment of RNA-seq reads from the GEO Dataset: GSE34550 [34] corresponding to mouse CD4+ T cells treated or not with CD3/CD28 antibodies. The alignment was performed with the UCSC genome browser on the mm10 assembly of the mouse genome and shows the peaks of the reads, the annotated gene model and the degree of sequence conservation among mammals.

**Figure 3 ncrna-08-00040-f003:**
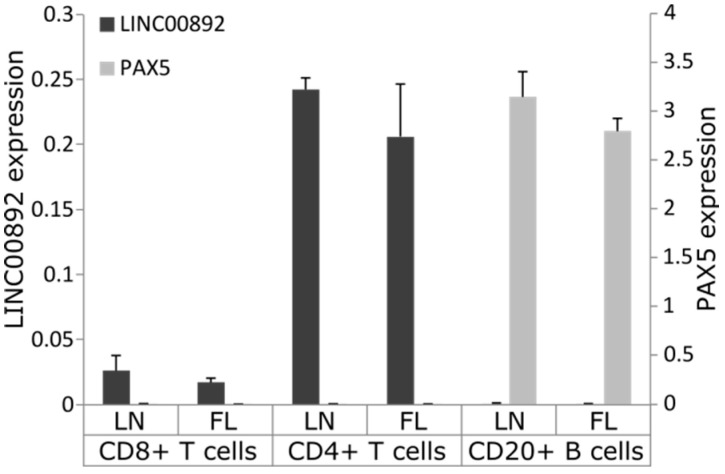
Expression analysis of LINC00892 and PAX5 in sorted cells. The analysis of gene expression of LINC00892 and PAX5 (as a B-cell marker) was performed using RT-qPCR from RNA extracted from CD4+ T cells, CD8+ T cells and CD20+ B cells sorted from a lymph node (LN) hyperplasia sample and an FL sample. The experiment has been performed in triplicate. GUSB was used as reference gene.

**Figure 4 ncrna-08-00040-f004:**
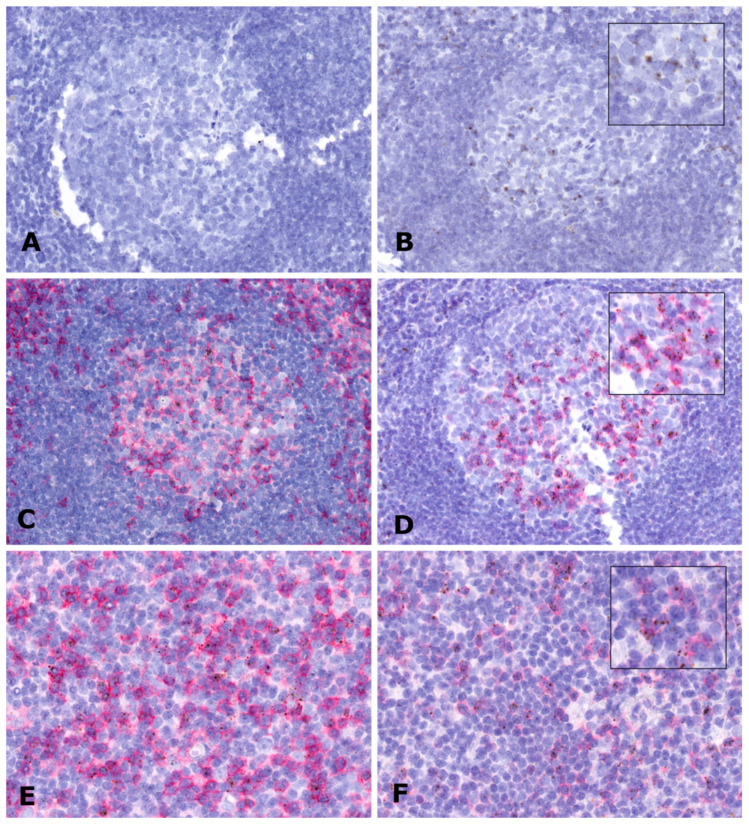
Intra-tissue LINC00892 localization using in situ hybridization (ISH) and immuno-histochemistry (IHC). RNA expression was analyzed by ISH using the RNAscope technology on FFPE slides from an LN (**A**–**D**) and an FL sample (**E**,**F**). Protein expression was investigated by IHC on FFPE slides from an LN (**C**,**D**) and an FL sample (**E**,**F**). LINC00892 expression is shown in brown. Expression of the proteins CD4 and PD1 is shown in red. (**A**) LN tissue stained by ISH with a negative control probe. (**B**) LN tissue stained by ISH with a LINC00892 probe. (**C**) LN tissue stained by ISH with a LINC00892 probe and by IHC with an anti-CD4 antibody. (**D**) LN tissue stained by ISH with a LINC00892 probe and by IHC with an anti-PD1 antibody. (**E**) FL tissue stained by ISH with a LINC00892 probe and by IHC with an anti-CD4 antibody. (**F**) FL tissue stained by ISH with a LINC00892 probe and by IHC with an anti-PD1 antibody.

**Figure 5 ncrna-08-00040-f005:**
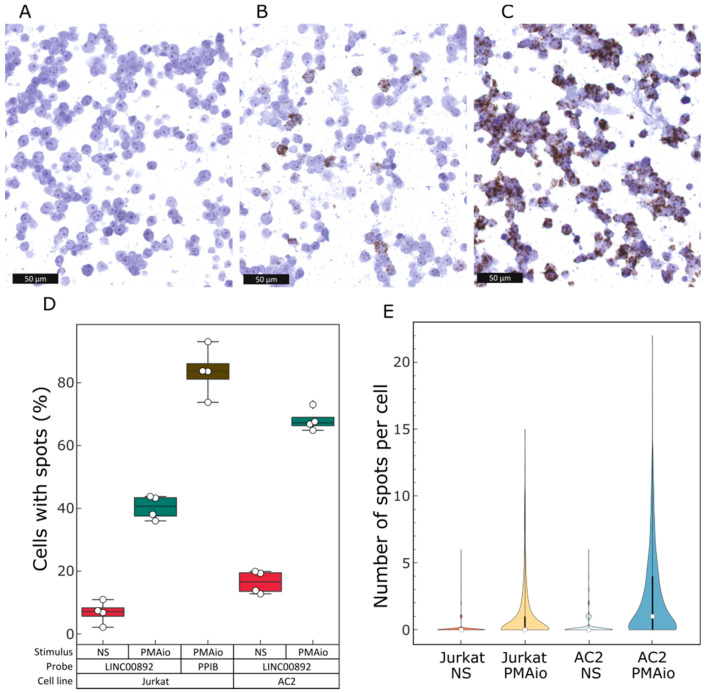
Single-cell analysis of LINC00892 expression in Jurkat cells. (**A**–**C**) In situ hybridization (ISH) analysis in non-stimulated Jurkat cells (**A**) or Jurkat cells stimulated with PMA and ionomycin and left growing for 48 h after stimulus withdrawal (**B**,**C**). Cells in (**A**,**B**) were stained with a probe specific for LINC00892. Cells in (**C**) were stained with a probe specific for cyclophilin B (PPIB). (**D**) Quantification of the number of cells having specific ISH spot signals in samples from parental Jurkat cells and from a sub-clone of the same Jurkat cells (AC2), using software for image analysis. For each condition, 4 different regions of interest (ROI) were analyzed. (**E**) Image analysis quantification of the number of spot per cell in samples from parental Jurkat cells and from a sub-clone of the same Jurkat cells (AC2).

**Figure 6 ncrna-08-00040-f006:**
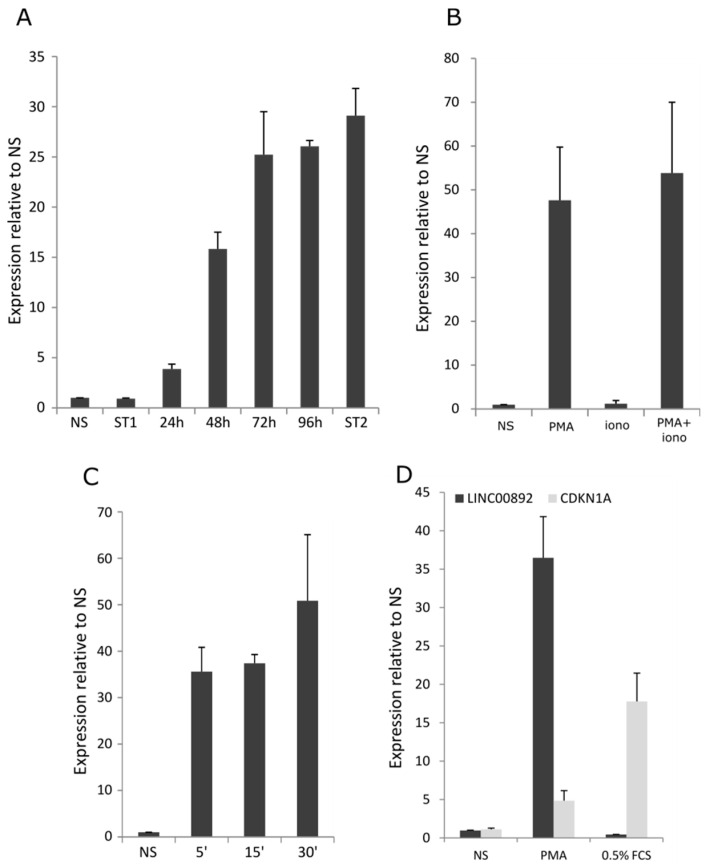
LINC00892 induction is late and persistent. (**A**) RT-qPCR analysis of LINC00892 expression in Jurkat cells left untreated (NS), stimulated with PMA/ionomycin for 3 h (ST1), washed and collected after the indicated hours after stimulation, and after a second 3 h PMA/ionomycin stimulus (ST2). (**B**) RT-qPCR analysis of LINC00892 expression in Jurkat cells left untreated (NS), stimulated with PMA only, ionomycin only or both PMA and ionomycin for 3 h and collected 4 days after the stimulus. (**C**) RT-qPCR analysis of LINC00892 expression in Jurkat cells left untreated (NS), stimulated with PMA only for 5, 15 or 30 min and collected 4 days after the stimulus. (**D**) RT-qPCR analysis of LINC00892 and CDKN1A expression in Jurkat cells left untreated (NS), stimulated with PMA only for 3 h and collected 2 days after the stimulus (PMA) or cultivated for 48 h in medium containing 0.5% FCS. Experiments in A, B and C were performed in triplicate. The experiment in D was performed in duplicate.

**Figure 7 ncrna-08-00040-f007:**
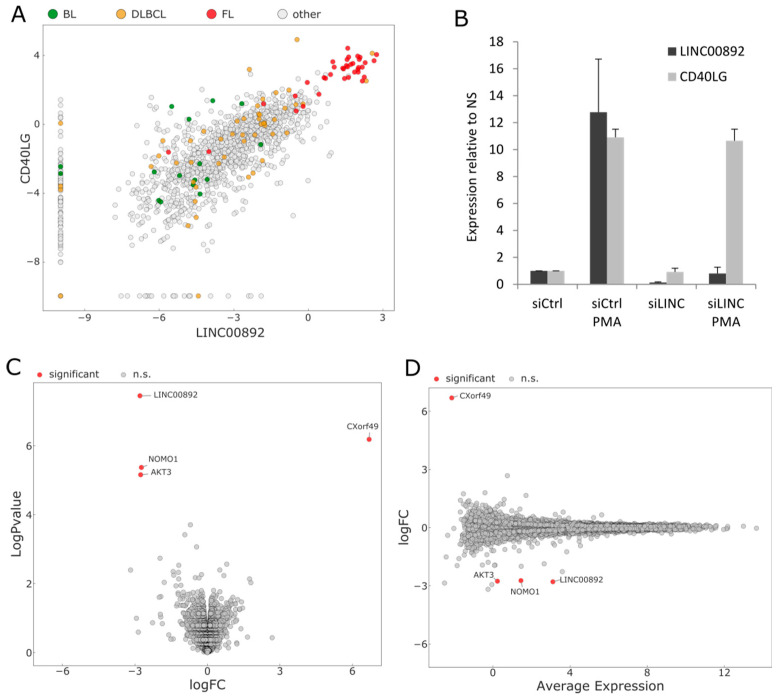
Expression correlation between LINC00892 and CD40LG genes. (**A**) Scatter plot analysis of the normalized expression levels of LINC00892 and CD40LG in the data from the Pan-Cancer Analysis of Whole Genomes (PCAWG) study. Expression values from lymphoma samples are marked with colors. (**B**) RT-qPCR analysis of LINC00892 and CD40LG expression in Jurkat cells transfected either with a non-targeting siRNA pool (siCtrl) or with an siRNA pool targeting LINC00892 (siLINC) in cells treated or not with PMA for 3 h. The experiment was performed in triplicate. (**C**) Volcano plot of RNA-seq data from Jurkat cells treated as in (**B**), comparing siLINC vs. siCtrl. (**D**) MD plot of RNA-seq data from Jurkat cells treated as in (**B**) comparing siLINC vs. siCtrl.

**Figure 8 ncrna-08-00040-f008:**
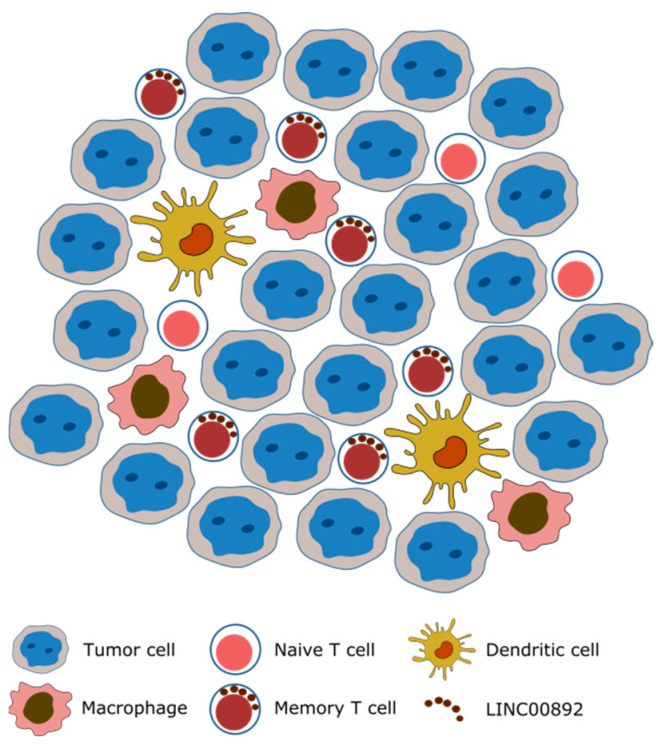
Possible use of LINC00892 as a biomarker for memory T cells: The figure proposes that in the analysis of a follicular lymphoma tissue, LINC00892 expression by ISH could help differentiate memory T cells from naïve T cells.

## Data Availability

Not applicable.

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
