# Peer review of "LINC00892 Is an lncRNA Induced by T Cell Activation and Expressed by Follicular Lymphoma-Resident T Helper Cells"

_ncrna, 2022, doi:10.3390/ncrna8030040_

Round 1
Reviewer 1 Report
Reviewer’s comment
ncrna-1713209
In the present article, the authors have characterized lncRNA LINC00892 as a potential biomarker for the detection of CD4+ memory T cells in both normal and tumor tissues. This article is very interesting. Cutting edge experiments were performed neatly. Overall the article reports excellent work dedicated to biologically important processes. This can be considered for publication in this international journals of repute.
Here are a few concerns
- Authors are advised to present a schematic in the discussion section to elucidate the function of LINC00892 as the biomarker for activated T cells.
- A scale bar will be useful for the image associated with Single-cell analysis (fig 5A-C)
Author Response
We are extremely thankful to the reviewers for the interest in our manuscript and the helpful comments provided. All the points raised by the reviewers were systematically addressed in this revision. Thanks to reviewers comments we believe the quality of the manuscript has clearly improved. A point-by-point response to reviewers’ comments is provided below.
Reviewer 1
In the present article, the authors have characterized lncRNA LINC00892 as a potential biomarker for the detection of CD4+ memory T cells in both normal and tumor tissues. This article is very interesting. Cutting edge experiments were performed neatly. Overall the article reports excellent work dedicated to biologically important processes. This can be considered for publication in this international journals of repute.
Here are a few concerns
1) Authors are advised to present a schematic in the discussion section to elucidate the function of LINC00892 as the biomarker for activated T cells.
2) A scale bar will be useful for the image associated with Single-cell analysis (fig 5A-C)
Our answer:
1) A graphic showing a possible function of LINC00892 as biomarker has been now added to the discussion as Figure 8.
2) A scale bar has been added to Figure 5.
Reviewer 2 Report
In this report, the authors describe a novel, spliced and conserved lncRNA, LINC00892, specific to the expression of T cells in an activate immune context and cancer. I really enjoyed reading this paper, as it responded gradually throughout reading to all my questions that came up. It is an extraordinarily comprehensive documentation of this novel lncRNA, and I am looking forward to future studies dissection the locus, and potentially other macromolecular binding partners of this lncRNA. I agree with the authors that this is outside of the scope of the manuscript. Oftentimes, adjacent lncRNA-protein coding loci are the result of (segmental) duplications or other duplications in the genome, and I'd be curious to know if the promoter regions are indeed co-regulated due to sequence similarity. In the future, it would also be important to test if the lncRNA has a function in the cytoplasm, as that is it's predominant location of expression.
My only suggesting is that authors make sure the statistical methods (number of experiments, testing, etc) are fully described and I would highly recommend to add this to the manuscript, and to make sure that a sufficient number of experiments is performed to ensure conclusive data. To complete, I would further suggest to study RNA stability (i.e. upon Transcriptional arrest) to potentially explain heterogeneity of expression in the stead state (i.e. if the transcript is highly unstable and gets produced only in well-spaced bursts of transcription could explain the high expression heterogeneity). In addition, does the protein level of CD40Lg also display high expression heterogeneity?
Author Response
We are extremely thankful to the reviewers for the interest in our manuscript and the helpful comments provided. All the points raised by the reviewers were systematically addressed in this revision. Thanks to reviewers comments we believe the quality of the manuscript has clearly improved. A point-by-point response to reviewers’ comments is provided below.
Reviewer 2
Comments and Suggestions for Authors
In this report, the authors describe a novel, spliced and conserved lncRNA, LINC00892, specific to the expression of T cells in an activate immune context and cancer. I really enjoyed reading this paper, as it responded gradually throughout reading to all my questions that came up. It is an extraordinarily comprehensive documentation of this novel lncRNA, and I am looking forward to future studies dissection the locus, and potentially other macromolecular binding partners of this lncRNA. I agree with the authors that this is outside of the scope of the manuscript. Oftentimes, adjacent lncRNA-protein coding loci are the result of (segmental) duplications or other duplications in the genome, and I'd be curious to know if the promoter regions are indeed co-regulated due to sequence similarity. In the future, it would also be important to test if the lncRNA has a function in the cytoplasm, as that is it's predominant location of expression.
My only suggesting is that authors make sure the statistical methods (number of experiments, testing, etc) are fully described and I would highly recommend to add this to the manuscript, and to make sure that a sufficient number of experiments is performed to ensure conclusive data. To complete, I would further suggest to study RNA stability (i.e. upon Transcriptional arrest) to potentially explain heterogeneity of expression in the stead state (i.e. if the transcript is highly unstable and gets produced only in well-spaced bursts of transcription could explain the high expression heterogeneity). In addition, does the protein level of CD40Lg also display high expression heterogeneity?
Our answer:
1) Reviewer 2 is right to point out that statistical methods and significance have not been described accurately in the manuscript. We have now edited both materials and methods and figure legends to address this issue.
2) We thank Reviewer 2 for suggesting possible explanations for the observed expression heterogeneity shown in Figure 5. Due to time limitations in the revision process, we have not been able to address this issue experimentally. Nevertheless, we have added a paragraph in the discussion of the manuscript, examining the issue in more details. The added paragraph is the following:
“Interestingly, single cell expression analysis showed that LINC00892 has a strong heterogeneous expression pattern in Jurkat cells (Figure 5). This observed heterogeneous expression in single cells of a clonal cell population, may be the resulting picture of asynchronous transcriptional bursts accompanied by rapid RNA degradation. Although we have not measured the half-life of the LINC00892 RNA, we did notice that upon ectopic expression in Jurkat cells, high levels of LINC00892 expression lasted only 24h (data not shown). On the other hand, a similar heterogeneous expression behavior has been observed previously in the analysis of cytokines mRNAs in cloned Th1, Th2 and Th0 cell lines [37]. Here the authors suggest that even in clonal T cell populations there may be a high level of plasticity in the expression of cytokine genes. In line with this hypothesis we find that a Jurkat sub-clone (AC2) is more homogeneus in its ability to increase LINC00892 expression (Figure 5D) and reaches higher expression levels. Interestingly clone AC2 has an enhanced ability to increase also CD40LG expression (data not shown) suggesting that single cell plasticity may be at the level of transcriptional regulation.”